# Recent Advances in Therapeutics for the Treatment of Alzheimer’s Disease

**DOI:** 10.3390/molecules29215131

**Published:** 2024-10-30

**Authors:** Amin Mahmood Thawabteh, Aseel Wasel Ghanem, Sara AbuMadi, Dania Thaher, Weam Jaghama, Donia Karaman, Rafik Karaman

**Affiliations:** 1Department of Chemistry, Birzeit University, Birzeit P.O. Box 14, West Bank, Palestine; athawabtah@birzeit.edu; 2Faculty of Pharmacy, Nursing and Health Professions, Birzeit University, Birzeit P.O. Box 14, West Bank, Palestine; aseel.wasel@gmail.com (A.W.G.); abumadisarah@gmail.com (S.A.); daniathaher2000@gmail.com (D.T.); weam_rezeq@yahoo.com (W.J.); 3Pharmaceutical Sciences Department, Faculty of Pharmacy, Al-Quds University, Jerusalem 20002, Palestine; kdonia65@yahoo.com; 4Department of Sciences, University of Basilicata, Via dell’Ateneo Lucano 10, 85100 Potenza, Italy

**Keywords:** anti-amyloid drugs, aducanumab, lecanemab, Alzheimer’s disease, chaperones, DMT

## Abstract

The most prevalent chronic neurodegenerative illness in the world is Alzheimer’s disease (AD). It results in mental symptoms including behavioral abnormalities and cognitive impairment, which have a substantial financial and psychological impact on the relatives of the patients. The review discusses various pathophysiological mechanisms contributing to AD, including amyloid beta, tau protein, inflammation, and other factors, while emphasizing the need for effective disease-modifying therapeutics that alter disease progression rather than merely alleviating symptoms. This review mainly covers medications that are now being studied in clinical trials or recently approved by the FDA that fall under the disease-modifying treatment (DMT) category, which alters the progression of the disease by targeting underlying biological mechanisms rather than merely alleviating symptoms. DMTs focus on improving patient outcomes by slowing cognitive decline, enhancing neuroprotection, and supporting neurogenesis. Additionally, the review covers amyloid-targeting therapies, tau-targeting therapies, neuroprotective therapies, and others. This evaluation specifically looked at studies on FDA-approved novel DMTs in Phase II or III development that were carried out between 2021 and 2024. A thorough review of the US government database identified clinical trials of biologics and small molecule drugs for 14 agents in Phase I, 34 in Phase II, and 11 in Phase III that might be completed by 2028.

## 1. Introduction

The pharmaceutical industry has been closely monitoring the advancement of AD therapy in an attempt to progress neurological disease treatment [1,2]. An intricate set of symptoms characterizes AD, a neurodegenerative disease that advances slowly. For this reason, there are still problems associated with this illness that scientists and physicians encounter globally [3,4]. Given the advancing age of the global population, there is a growing demand for effective medicines such as DMTs that have emerged as a promising treatment option. These novel strategies aim to both drastically alter the illness’s trajectory and cure its symptoms, which might make a big difference in the battle against this debilitating condition [5,6].

### 1.1. Pathophysiological Mechanisms and Symptoms of AD

AD is a progressive neurodegenerative disorder and the leading cause of dementia among older adults, characterized by a gradual decline in cognitive function, memory loss, and behavioral changes, ultimately leading to complete dependence on caregivers [7].

The pathophysiological process of AD is multifaceted, involving several key hypotheses: the cholinergic, glutamatergic, amyloid, tau protein, inflammatory, microbiota–gut–brain axis, oxidative stress, metal ion, and abnormal autophagy hypotheses, as shown in the summary of AD pathophysiology in Figure 1 [8]. The cholinergic hypothesis posits that the degeneration of cholinergic neurons, particularly in the basal forebrain, is a central feature of AD and is closely linked to cognitive decline. This hypothesis suggests that a reduction in the synthesis and release of ACh impairs its physiological functions, which are critical for learning, memory, and other cognitive processes. The decline in choline acetyltransferase (ChAT) activity, combined with the detrimental effects of amyloid beta (Aβ) on cholinergic signaling, leads to decreased ACh levels. Consequently, this reduction contributes to the cognitive deficits observed in AD patients. AChE inhibitors, such as donepezil, rivastigmine, and galantamine, have been developed as treatments based on this hypothesis, although their efficacy is limited and they do not halt disease progression [7,9].

The glutamatergic hypothesis posits that dysregulation of glutamate, the main excitatory neurotransmitter in the central nervous system, plays a critical role in the pathogenesis of AD. In AD, there is an over-stimulation of NMDA receptors, which leads to an excessive influx of sodium and calcium ions into neurons. This dysregulation can cause neuronal swelling and initiate various calcium-dependent processes, including the production of reactive oxygen species (ROS), disruption of mitochondrial function, and activation of necrotic or apoptotic pathways, ultimately resulting in excitotoxic damage to neurons [10].

The amyloid hypothesis emphasizes that the accumulation of Aβ plaques in the brain is a central event in the pathogenesis of AD, leading to neurodegeneration and cognitive decline. Aβ originates from the processing of the amyloid precursor protein (APP) through sequential cleavage by β-secretase and γ-secretase, resulting in various lengths of Aβ fragments, with Aβ40 and Aβ42 being predominant. The hydrophobic C-terminus of Aβ42 facilitates its aggregation and the formation of senile plaques. Mutations in presenilin-1 (PS1) can promote Aβ accumulation through increased production and impaired autophagy functions. However, the relationship between FAD mutations and Aβ levels is complex, and the plaque formation in sporadic AD is influenced by a dynamic imbalance between Aβ production and clearance mechanisms [11,12].

The tau protein hypothesis focuses on the hyperphosphorylation of tau protein leading to its detachment from microtubules, resulting in the formation of neurofibrillary tangles (NFTs) which are a hallmark of AD. Under normal conditions, tau stabilizes microtubules, but pathological conditions disrupt the balance of kinases and phosphatases, causing tau to become hyperphosphorylated. This hyperphosphorylation promotes tau aggregation into paired helical filaments and NFTs, which correlate strongly with clinical symptoms of AD. Additionally, tau pathology is associated with neuroinflammation, as pro-inflammatory cytokines can increase tau transcription and activate inflammatory pathways that further exacerbate tau-related neurodegeneration [13,14].

The inflammatory hypothesis suggests that chronic neuroinflammation plays a significant role in the pathogenesis of AD. This hypothesis posits that various pathological factors, such as Aβ plaques, pro-inflammatory cytokines, and oxidative stress, activate microglia, the central nervous system’s innate immune cells. Upon activation, microglia undergo morphological changes and initiate a variety of inflammatory responses, which can lead to the release of pro-inflammatory mediators that exacerbate neurotoxicity. The persistent activation of microglia may result in a failure to resolve inflammation, contributing to neuronal damage and the cognitive decline associated with AD [15,16].

The microbiota–gut–brain axis hypothesis suggests a bidirectional communication system between the gut and the brain, involving metabolic, endocrine, neural, and immune pathways. Dysbiosis, or disruption of the microbiota’s composition and functionality, can occur due to factors such as dietary changes, antibiotic use, psychosocial stress, or immune irregularities, leading to a compromised intestinal epithelial barrier. This allows harmful substances and microorganisms to enter the bloodstream, triggering systemic inflammation. Such inflammation may enable inflammatory mediators to cross the blood–brain barrier (BBB) and affect microglia, exacerbating neuroinflammation, which ultimately contributes to neuronal degeneration and damage [17,18].

The oxidative stress hypothesis indicates that oxidative stress plays a crucial role in the pathogenesis of AD. It suggests that an imbalance between the production of reactive oxygen species (ROS) and the body’s ability to detoxify these harmful compounds leads to damage of neuronal membrane lipids, proteins, and nucleic acids, ultimately causing neuronal cell death. Key contributors to oxidative stress in AD include mitochondrial dysfunction, which reduces the activities of essential enzymes and disrupts mitochondrial dynamics. This oxidative stress is interconnected with other pathological processes in AD, such as the modulation of APP processing and tau phosphorylation, which leads to the formation of neurofibrillary tangles. Additionally, the presence of free metals and complexes of Aβ can catalyze ROS production, further exacerbating neuronal damage [19,20].

The metal ion hypothesis examines the dysregulation of metal ions, particularly copper, zinc, and iron, which plays a significant role in the pathogenesis of AD. This hypothesis suggests that metal dyshomeostasis can lead to oxidative stress, which is linked to various pathological events in AD, such as amyloidosis and tauopathy. For instance, iron-induced oxidative stress can result in the release of iron from iron-containing proteins, leading to neuronal death through mechanisms like ferroptosis and lipid peroxidation. Similarly, copper can induce toxicity by binding to specific proteins, resulting in cellular damage. Zinc deficiency is also highlighted as a factor that may contribute to glutamate excitotoxicity and synaptic dysfunction in AD. Overall, the accumulation and mismanagement of metal ions are closely associated with the neurodegenerative processes observed in AD [21].

The abnormal autophagy hypothesis suggests that disruptions in the autophagy process are closely linked to the pathogenesis of AD. Autophagy is essential for maintaining cellular homeostasis by degrading and recycling intracellular protein aggregates and damaged organelles. In AD, autophagy defects can lead to the accumulation of Aβ and the abnormal aggregation of tau protein, contributing to the disruption of protein homeostasis networks. These defects may arise from genetic factors, reduced expression of related proteins, and impaired vesicular transportation, ultimately resulting in the accumulation of damaged organelles, such as dysfunctional mitochondria [22,23].

The most common symptoms of AD are short-term memory loss, which is followed by a progressive impairment in both implicit and explicit memory. Those who have the condition also exhibit neuropsychiatric symptoms, and with time, they lose their ability to perform basic daily duties [24,25,26,27,28]. AD is categorized into three stages: mild, moderate, and severe [24,25,26,27,28]. The signs and symptoms of AD in each of its three stages are included in Table 1.

Patients with AD and their families, as well as the healthcare system, bear a heavy burden. Families must deal with social, economic, and emotional difficulties as the illness worsens. The emotional toll is high, causing worry, despair, and exhaustion in those who provide care [35,36].

### 1.2. Causes and Factors of AD

Understanding the fundamental causes of AD is crucial to developing effective medicines and interventions. The various causes of AD will be looked at in this review, with a focus on inherited, environmental, and lifestyle factors [29,37,38].

One of the main variables that causes AD is genetics. Certain genes that are inherited have been connected to the condition as risk factors. The two main gene categories associated with AD are deterministic and risk-factor genes [29,37]. Deterministic genes are not always the direct cause of AD. The most well-known genes are those for presenilin-1 (PSEN1), presenilin-2 (PSEN2), and amyloid precursor protein (APP). Mutations in these genes can cause early-onset familial AD, which often manifests before the age of 65 [38].

According to studies, having these gene mutations practically guarantees that a person will develop the sickness [39,40,41]. The most well-known risk-factor gene is apolipoprotein E (APOE). The AD risk is greatly increased by the APOE ε4 allele. A person with one copy of this allele is more likely to be affected than a person with two copies [42,43,44].

AD has been linked to a number of environmental variables in addition to genetic predispositions [37]. Dementia risk has been shown to rise with toxic exposure to heavy metals as lead, mercury, and aluminum [45,46,47,48]. Moreover, exposure to air pollution may worsen neuroinflammation and cognitive decline [49,50]. Furthermore, several studies have connected lifestyle choices to a higher risk of AD [51]. AD risk factors include obesity, diabetes, high blood pressure, and high cholesterol [51,52,53].

Furthermore, sedentary lifestyles and bad dietary habits, such as consuming large amounts of carbs and saturated fats, have been connected to an increased risk [54,55,56,57]. Conversely, maintaining cardiovascular health, eating a balanced diet rich in antioxidants, and exercising frequently can all lower the incidence of AD [55,58]. Figure 2 summarizes the causes and factors of AD.

## 2. Disease-Modifying Therapeutics

The concept of disease-modifying treatments, or DMTs, has gained a lot of momentum in the medical community, particularly in the treatment of chronic illnesses like AD and Multiple Sclerosis (MS). DMTs are treatments that postpone, moderate, or even reverse the evolution of an illness by addressing its underlying cause. They are not the same as symptomatic therapies, which focus on the symptoms of a disease without addressing its root cause [6,59,60].

### 2.1. Definition of DMT

DMTs are a family of medications designed to change the course of a disease rather than only cure its symptoms. DMTs consist of three primary components: therapy, sickness, and modification [37,59]. DMTs are intended to delay the progression of long-term conditions such as AD and ultimately improve the patient’s quality of life [60].

Based on their mechanisms of action, these treatments can be divided into several categories, including immunomodulatory medications, neuroprotective medications, and others that target specific pathways connected to disease processes [61,62,63].

While symptomatic therapies may provide immediate relief from pain or fatigue, DMTs specifically work to change the immune response or repair brain damage [64,65]. This shift in focus is crucial for treating progressive illnesses because it addresses both the disease’s symptoms and the underlying issues that cause it to worsen [61,65,66,67].

### 2.2. Purpose of DMT

The primary goal of DMTs for AD patients is to slow the rate of cognitive decline [66,67,68]. These therapies aim to preserve neuronal connectivity and function by concentrating on specific pathological pathways [69,70,71]. Amyloid-beta buildup in the brain is believed to be crucial for the onset and progression of AD, and certain DMTs target this accumulation [70,71,72].

Lowering amyloid levels may help to lessen the disease’s related neurodegeneration and cognitive impairment, according to research [73,74]. Enhancing neuroprotection is one of DMTs’ main objectives [74]. This entails protecting neurons from oxidative stress and inflammation, among other potential causes of injury [73,74,75]. Neuroprotective drugs may be able to postpone the development of more severe symptoms by preserving the health and function of neurons [76,77]. Furthermore, some DMTs work to support neurogenesis, the process that results in the formation of new neurons, which will support cognitive function even as the disease advances [78]. Moreover, DMTs work to prolong patients’ independence as much as possible in order to enhance their general quality of life [79,80,81].

These treatments can help patients to maintain their everyday functioning and cognitive capacities by altering the course of the disease, enabling them to participate more fully in their lives and communities [60]. Since amyloid-beta (Aβ) plaques and tau tangles are characteristic of AD and are thought to be important in the onset and development of the disease, the main focus of these therapies has been on removing or reducing them. Research indicates that lowering amyloid levels may help to mitigate the neurodegeneration and cognitive impairment associated with the disease [82,83,84].

### 2.3. Mechanisms of Action

The development of amyloid-beta (Aβ) plaques, neuroinflammation, hyperphosphorylation of tau protein resulting in neurofibrillary tangles, and synaptic dysfunction are among the pathophysiological features of AD. DMT does more than merely treat symptoms; it aims to change the fundamental path of the disease. DMTs use a variety of processes to accomplish their objectives; these mechanisms are intricate and varied. Based on how they work, DMTs can be divided into a number of categories [37,76,83].

#### 2.3.1. Amyloid Targeting Therapies

The focus of efforts to treat AD, a neurodegenerative condition marked by progressive cognitive deterioration, has shifted to amyloid-targeting medicines. The main goal of these treatments is to lessen or completely eradicate beta-amyloid (Aβ) plaques in the brain, which are thought to be a major factor in the development of AD [85,86,87].

A protein called amyloid beta collects in the brains of people who have AD to create plaques. It is believed that the buildup of these plaques sets off a series of neurological events that eventually result in cognitive loss. Studies have shown that Aβ buildup can start 15–20 years before clinical symptoms appear, indicating that therapy effectiveness may depend on early intervention [88,89]. According to the amyloid cascade hypothesis, tau protein tangles and neuroinflammation are brought on by Aβ aggregation, which in turn leads to neuronal death and cognitive decline [90,91,92].

Even with the theoretical underpinning that the amyloid hypothesis offered, there have been several obstacles in the way of turning this information into practical treatments. Several clinical investigations focusing on Aβ have produced contradictory findings, casting doubt on the veracity of the amyloid theory in general. While lowering Aβ levels can decrease cognitive deterioration, not all patients will definitely see improved clinical outcomes from this strategy, according to certain research [86,93]. This intricacy emphasizes the necessity of a comprehensive comprehension of Aβ’s function in AD and the processes through which amyloid-targeting treatments function. Based on the amyloid cascade theory. The amyloid-targeting therapy medications are discussed in Section 3. The principal Aβ-targeted therapies’ method of action is illustrated in Figure 3.

#### 2.3.2. Tau-Targeting Therapies

In the effort to create potent AD therapeutics, tau-targeting medicines have taken center stage. In contrast to the amyloid-beta (Aβ) plaques that are the subject of more extensive research, tau protein abnormalities, like hyperphosphorylation and aggregation, are becoming more well acknowledged for their important role in the development of AD. Tau is a protein that is associated with microtubules and is essential for preserving the structure and functionality of neurons. Tau maintains microtubules in healthy neurons, which are necessary for intracellular transport. However, tau experiences pathogenic alterations in AD, such as hyperphosphorylation, which results in the development of neurofibrillary tangles. Given their strong correlation with cognitive decline and neurodegeneration, tau is an attractive target for therapeutic intervention [13,14].

Research has indicated that tau pathology has a stronger correlation with cognitive deficits than Aβ plaques, indicating that treatment benefits from targeting tau may outweigh those from concentrating only on amyloid clearance [94,95]. Over time, our knowledge of tau’s function in AD has changed dramatically. The focus of early studies was on the aggregation mechanism of tau and its post-translational alterations. Nevertheless, as the complexities of tau disease became more apparent, scientists started looking into other ways to prevent tau aggregation and support tau’s healthy operation. This change in emphasis has made room for novel treatment strategies meant to lessen the consequences of tau-related neurodegeneration [13,95].

In clinical trials, immunotherapies make up the majority of tau-targeting treatments. The aim of these treatments is to trigger an immunological reaction against tau proteins, which will help to remove them from the brain. Passive immunization with monoclonal antibodies and active immunization techniques that encourage the body’s immune system to generate its own antibodies against tau are two examples of different immunotherapeutic treatments [96,97].

Preclinical research has indicated the potential for multiple immunotherapies that target tau. In animal models, case in point, antibodies that bind selectively to phosphorylated tau have shown promise in lowering tau pathology. Numerous clinical trials aimed at assessing the safety and effectiveness of these medicines in humans have been prompted by these discoveries. These treatments aim to directly target tau, which should slow or even stop the progression of AD. Apart from immunotherapies, alternative therapeutic approaches have also been investigated. These include substances that stabilize microtubules to return tau function to normal and tiny molecules that prevent tau from aggregating. But a lot of these methods have had difficulties in the clinic, frequently because of problems with toxicity or ineffectiveness [95,98]. Based on the tau-targeting theory. The principal tau-targeting therapies’ route of action is shown in Figure 4.

#### 2.3.3. Neuroprotective Therapies

It is possible to overexpress heat shock proteins (HSPs), which are neuroprotective agents. A molecular chaperone is a protein, such as HSPs, that aids in the folding or unfolding of other non-native proteins. The majority of neurodegenerative illnesses, including AD, are primarily caused by cell death, which is brought on by misfolding and aggregating proteins. Molecular chaperones can be classified as either external (clustered proteins, such alpha-macroglobulin and clustering) or intracellular (heat shock proteins, such as Hsp40, Hsp60, Hsp70, Hsp90, Hsp100, and Hsp110) [37,99,100]. Protein folding depends on HSPs, which also protect cells from harmful stress-related events. The following are the two groups of HSPs: (a) classic HSPs, whose molecular weight of ATP-binding sites is at least 60 kDa. Hsp100, Hsp90, Hsp70, and Hsp60 are members of this family, in addition to (b) the tiny HSPs with a molecular weight of 40 kDa or less, which include αB-crystalline, Hsp27, Hsp20, HspB8, and HspB2/B3. These proteins are able to assist other HSPs in their function of refolding. Inadequate operation of these systems can lead to oxidative stress, mitochondrial dysfunction, and various other pathological conditions that damage the body, cause neuronal death, and exacerbate dementia. Amyloidogenic proteins (Aβ) and tau are two examples of heat-shock proteins (HSPs) that can both promote and inhibit the aggregation of misfolded proteins [37,101].

## 3. AD Current Treatment Drugs

According to the theory that AD results from a reduction in the production of acetylcholine (ACh), blocking the enzyme acetylcholinesterase, which breaks down acetylcholine in synapses, will raise cholinergic levels. Without affecting the disease’s course, this leads to a persistent build-up of acetylcholine and the activation of cholinergic receptors, which enhances cognitive and brain cell performance. Donepezil, galantamine, and rivastigmine are the cholinesterase inhibitors that are most frequently prescribed [102,103,104,105,106].

The second generation of AChEIs, donepezil (labeled as **1**, Figure 5), is an indanonebenzylpiperidine derivative that is authorized for the treatment of AD in all phases. By reversibly binding to acetylcholinesterase, it slows the hydrolysis of acetylcholine, increasing the concentration of ACh at synapses [106,107]. The gastrointestinal and neurological systems are the subjects of modest, temporary cholinergic side effects that are well-tolerated with donepezil [106,107,108,109]. With a dual mode of action, galantamine (labeled as **2**, Figure 5) is a selective tertiary isoquinoline alkaloid that may both bind allosterically to the α-subunit of nicotinic acetylcholine receptors and activate them, as well as competitively inhibit AChE [110,111]. It is a first-line medication for mild-to-severe AD that has a good efficacy and tolerability in improving behavioral symptoms, activities of daily living, and cognitive performance [110,112,113]. ACh metabolism is inhibited by rivastigmine (**3**, Figure 5), a pseudo-irreversible inhibitor of AChE and butyrylcholinesterase (BuChE) [114,115,116]. It is referred to as pseudo-irreversible because it dissociates more slowly than AChE. AChE and BuChE metabolize it at the synapse [115,116]. It is authorized for use in mild-to-moderate Parkinson’s disease dementia and mild-to-moderate AD [116,117,118].

Even if the medicine becomes more tolerated with time, oral delivery is linked to unfavorable effects that affect adherence. Nausea, vomiting, dyspepsia, asthenia, anorexia, and weight loss are some of these side effects. Compared to oral pills, transdermal patches are more pleasant and provide a smaller dosage, which leads to fewer side effects. Since AD patients have memory and swallowing issues that make it difficult for them to take oral medications as prescribed, transdermal patches are the best way to administer rivastigmine to these patients [119,120,121].

The neurotoxicity seen in AD patients is caused by the glutaminergic system being overactivated, which is inhibited by memantine (labeled as **4**, Figure 5), an NMDAR low-affinity uncompetitive antagonist [37,122,123,124]. Memantine is safe and well-tolerated; it is used to treat moderate-to-severe AD alone or in combination with AChEI [124,125,126]. For moderate-to-severe AD dementia, a combination of a glutamate regulator (memantine) and a cholinesterase inhibitor (donepezil) is approved [127,128,129]. Constipation, headache, nausea, vomiting, lack of appetite, increased frequency of bowel movements, dizziness, disorientation, and headaches are among the side effects [127,128,129].

The first disease-modifying medication for AD is a recombinant monoclonal antibody called aducanumab, also known by the brand name Aduhelm. It targets amyloid beta specifically [37]. It functions by lowering the quantity of amyloid deposits or brain plaques [130,131]. Aducanumab should only be explored for people with moderate AD or mild cognitive impairment (MCI) resulting from AD itself. Every four weeks, an intravenous (IV) dose is administered [132,133]. Aducanumab has been given FDA approval under the expedited approval pathway to treat moderate AD [134,135,136]. The reduction in amyloid-beta brain plaques, a purported surrogate endpoint that indicates the “progression” of AD, and the encouraging clinical outcomes in one of the two pivotal Phase III trials were two of the factors cited for aducanumab’s expedited approval [130,134,135,136].

Lecanemab (LEQEMBI, brand name), a different monoclonal antibody known as humanized immunoglobulin G1 (IgG1), which specifically targets aggregated forms of amyloid beta, was also given fast approval by the FDA earlier in the year [134,135,136]. Patients who are at a suitable stage of the disease as when treatment was started in clinical trials—mild cognitive impairment/mild dementia—should begin LEQEMBI. There is insufficient safety or efficacy information on starting treatment sooner or later than in the conducted studies [137,138,139].

### 3.1. Oral Immunotherapies or Small Molecules Containing DMTs

Amyloid-beta (Aβ) and tau proteins, which are essential to the pathophysiology of the illness, are the usual targets of DMT for AD. A number of immune therapies, such as AN-1792, a synthetic Aβ peptide (human Aβ1-42 peptide with immune adjuvant QS-21), have been developed and tested for the treatment of AD. However, AN-1792 was withdrawn during Phase II of the trial due to side effects associated with meningoencephalitis, which were reported in 6% of patients [140,141,142,143,144]. Clinical trials have investigated many anti-Aβ antibodies, such as solanezumab [145,146], bapineuzumab [147,148,149], and aducanuamb. Additionally, gantenerumab [150,151,152] and crenezumab [92,147,149] were examined. These are intended to target brain amyloid-beta plaques, but their efficacy has been met with a number of obstacles. CAD106, a further active Aβ immunotherapy, is presently undergoing clinical development. It involves administering multiple doses of the whole Aβ1–6 peptide linked to the ADP-Q carrier, a virus-like particle that generates high-titer anti-Aβ antibodies [140,141,153,154].

Most of the small compounds used in AD therapy are directed towards important enzymes and pathways that are involved in the processing of tau or amyloid-beta proteins. This includes γ-secretase inhibitors, such as tarenflurbil (**7**, Figure 5) [155,156], avagacestat (labeled as **6**, Figure 5) [157,158], and semagacestat (**5**, Figure 5) [159,160].

Their purpose is to lower the synthesis of amyloid beta by preventing the gamma secretase enzyme from functioning. In order to decrease the synthesis of amyloid beta, β-secretase inhibitors (BACE inhibitors) such lanabecestat (labeled as **8**, Figure 5) [157,161,162], verubecestat (labeled as **9**, Figure 5) [157,163,164], and atabecestat (labeled as **10**, Figure 5) target the enzyme Bace-1 [157,165]. Similar to other BACE-1 inhibitors, CNP520 (umibecestat (labeled as **11**, Figure 5)) has been shown to reduce the burden of amyloid-beta plaques in preclinical animals; it is presently undergoing testing phases [157,166,167].

The purpose of the second class of tiny compounds, known as tau aggregation inhibitors, is to stop or interfere with the development of harmful tau aggregates. Due to problems with binding efficiency, which show up as blue discolorations in urine, methylene blue (labeled as **12**, Figure 5), which was also tried in Phase II clinical trials for mild-to-moderate AD (for suppressing tau aggregation), has drawn a lot of criticism [168,169,170].

Tideglusib (NP-031112 (labeled as **13**, Figure 5)), an inhibitor of GSK3β, targets tau hyperphosphorylation, which is required for the development of tau tangles [171,172,173]. The tyrosine kinase inhibitor saracatinib (AZD0530 (labeled as **14**, Figure 5)) is undergoing Phase II trials [174,175,176] and has been demonstrated to improve memory in transgenic mice models of AD. When taken as a whole, these tiny compounds and immunotherapies offer a more comprehensive strategy for changing AD by focusing on many pathogenic processes that are involved in the illness [37].

PDE4 inhibitors, which are oral small compounds that selectively inhibit PDE4, an enzyme involved in controlling cAMP levels, include roflumilast (**15**, Figure 5), which may lessen inflammation and increase neuronal survival [177,178,179,180]. Scientists are beginning to investigate whether they could be helpful for neurological illnesses, even though they have primarily been studied for conditions including asthma and chronic obstructive pulmonary disease (COPD) [181].

### 3.2. Recent DMT’s Clinical Trials

The most recent list of studies was provided by the 2024 AD drug development pipeline, which includes 164 trials (of which 127 are exploring possible treatments or medicines). There were slightly fewer trials overall than the year before. Of these trials, 41% look into disease-modifying small molecule medications, 34% look into biological agents that change disease, and the remaining trials focus on treatments that improve cognitive function or address neuropsychiatric symptoms linked to ADs [68,70,71,182,183,184]. Among the many pharmacological targets investigated in these studies are synaptic plasticity, neurotransmitter receptors, inflammation, and amyloid.

The process of creating medications to treat AD involves several possible clinical outcomes in various stages of the clinical process. In Phase I, there are 26 trials assessing 25 agents. Phase II trials (90 studies, 81 medicines): these trials continue to investigate the drugs’ safety and efficacy. Phase III trials, including 48 trials that tested 32 medications [37,183,184].

Aducanumab demonstrated inconsistent outcomes in its two main clinical trials, the Phase III EMERGE and ENGAGE research investigations. In the EMERGE experiment, when participants with AD were at risk for cognitive deterioration, there was a decrease in the amount of amyloid-beta plaques. But ENGAGE did not demonstrate the same effectiveness. In spite of this, the FDA granted expedited approval on the grounds of the EMERGE trial’s documented decrease in amyloid levels [132,185,186,187]. Lecanemab’s capacity to lower the amount of amyloid in the brain, as shown by PET scans, has led to its submission for expedited approval for the treatment of AD. After 79 weeks, lecanemab was found to have a lower total brain amyloid plaque burden than a placebo (Study 201). In particular, a mean difference of −73.5 centiloides and −0.31 standardized uptake value ratio (SUVR) units was found between the lecanemab and placebo treatment groups. Additionally, the treatment impact increased in a dose- and exposure-dependent manner; a 25% AUC therapy was probably superior to a placebo by >0.64. The study also suggested at week 79 that a risk reduction of 20% to 40% be applied to the advancement of clinical symptoms [187,188].

The effectiveness and safety of several monoclonal antibodies used to treat AD were assessed by meta-analyses of monoclonal antibodies. While both aducanumab and lecanemab slowed cognitive decline and reduced amyloid plaques, they were more likely to cause amyloid-related imaging abnormalities (ARIAs), which are thought to be the most significant side effect of both drugs. Aducanumab in particular was more likely to cause ARIA-E (edema) [37,92].

The US government database of clinical research studies and information, information from which is shown in Table 2 and Table 3, summarizes the clinical trials of disease-modifying biological agents and the clinical trials of disease-modifying small molecule drugs, respectively. These tables represent agents in the three phases of disease-modifying agents’ development for AD [38,46,68,70,71,157,182,183,189].

Challenges in AD Therapeutic Development

Disease heterogeneity

The interplay between phenotypes and genotypes is part of the heterogeneity of AD. Cognitive, neurological, psychological, and functional deficits are complicated and different as a result of these interactions with environmental factors. More and more research is recognizing the etiological and clinical heterogeneity of AD as a hallmark of AD and related diseases (ADRDs), which makes it more difficult to diagnose, treat, and develop and test new medications for AD. More work has to be conducted to take this variability into account in order to improve the results of AD clinical trials [190,191,192].

Blood–brain barrier penetration

An important impediment to the development of AD medications is the blood–brain barrier (BBB), which prevents potential drug candidates from entering the brain parenchyma. The BBB is impermeable to 98% of small molecules and around 100% of large molecules in relevant quantities. For a medicine to penetrate the blood–brain barrier, its physicochemical properties—such as its polar surface area, molecular weight, and lipophilicity as assessed by hydrogen bonding ability—are essential. A molecular weight less than 450 Daltons, a polar surface area less than 90 Å2, and a hydrogen bond donor number less than 3 are characteristics of candidates for BBB penetration. The indicated BBB penetration values are inconsistent with the physicochemical characteristics of monoclonal antibodies (solanezumab, aducanumab, bapineuzumab) or gamma-secretase inhibitors (semagacestat, tarenflurbil) [193,194,195].

Biomarkers and patient selection

AD clinical trials must have larger trial numbers and may include trial subpopulations referred to as “slow/no progressors” since they show little or no progression of the disease over time. This phenomenon is most likely caused by uncertainties in AD pathophysiology, such as the lack of validated diagnostic biomarkers and the low specificity of clinical diagnosis. This has been made worse by the increased focus given to the predementia stages of the disease: (1) asymptomatic predementia, also referred to as the preclinical phase or Stage 1–2, and (2) symptomatic predementia, also known as amnestic MCI or Stage 3.

Regulatory challenges

One of the regulatory obstacles in creating DMTs for AD is the insufficient knowledge and ability of regional regulatory organizations, in addition to post-study medication administration and sustained treatment supply. Moreover, there are several review levels with conflicting results and different guidelines for participant reimbursement for trial-related injuries. Moreover, biobanking and repository research are being scrutinized and restricted more harshly [196,197].

## 4. Future Directions and Emerging Therapies in DMT

Novel targets for DMT

The accumulation of tau protein in neurofibrillary tangles, which is known as tau protein modulation, provides a novel venue for disease modification. The goal of tau treatments is to prevent tau from aggregating and spreading. Numerous approaches, such as tau aggregation inhibitors, are being studied in an effort to address the tau protein problem and prevent tau tangle formation. Techniques for tau immunotherapy, such as those being tried with AADvac1, function by teaching the immune system to eliminate aberrant tau proteins from the brain. First, there is evidence that tau vaccinations may slow the disease’s course by reducing tau protein accumulation [198,199,200,201].

Inflammation, or neuroinflammation, is a crucial factor in the mechanisms behind AD. In the healthy aging human brain, amyloid-β plaques have the ability to cause persistent inflammation in microglia, which are the brain’s resident immune cells. TREM2 receptor interaction plays a key role in identifying microglia for the build-up of debris and damage during the regulation of microglial activation. TREM2 agonists are being investigated as potential therapeutic agents since they reduce inflammation and encourage amyloid-beta microglial phagocytosis [202,203,204,205].

Research on cytokine inhibition has focused on anti-inflammatory medications that interact with pro-inflammatory cytokines such as TNF-a and IL-1 beta. Treatment options for neuroinflammation associated with AD have been investigated, including IL-1β antagonists such as canakinumab [206,207,208]. Regarding the plasticity of synapses and neurodefense, it has been found that the decline in cognitive abilities in ADs is positively correlated with the breakdown of synaptic connections. Consequently, neurotransmitter modulation is useful. Cholinesterase inhibitors and NMDA receptor antagonists are the only two categories of medications that are prescribed to address symptoms [208].

Even more targeted strategies are being developed to enhance neurotransmitter activity. Examples of medications that may improve synaptic function include those that allosterically increase the M1 muscarinic acetylcholine receptor [209,210]. In order to preserve microtubules and maintain neuronal function, neuroprotective agents—substances such as NAP, the activity-dependent neuroprotective protein generated from ADNP—have been shown to be useful in preventing the death of neurons [211,212].

Precision Medicine Approaches

For ADs, precision medicine focuses on genetic and biomarker analysis to improve therapy approaches, risk assessment, and diagnosis. PSEN1 and PSEN2 mutations, which are closely associated with early-onset AD and impact the γ-secretase cleavage products, such as Aβ, are important genetic factors. These mutations have been linked to a high frequency of seizures, accounting for about 32% of cases. They may also be the cause of neuroinflammation by activating microglia and producing inflammatory cytokines. Furthermore, several risk loci associated with apoptosis and Wnt/β-catenin signaling pathways, including USP6NL/ECHDC3, PFDN1/HBEGF, BZRAP1-AS1, and TPBG, have been identified by genome-wide association studies (GWAS) [37,213,214].

The field of biomarkers is dominated by blood-based markers, including heart-type fatty acid-binding protein (HFABP), neurofilament light chain (NFL), neuron-specific enolase (NSE), total tau (t-Tau), phosphorylated tau (p-Tau), and monocyte chemoattractant protein-1 (MCP-1). These markers provide information about the underlying pathophysiology of AD and are crucial for detecting the disease, especially in those who have memory problems. With the use of fluorodeoxyglucose (FDG)-PET, magnetic resonance imaging (MRI), and positron emission tomography (PET) imaging for the detection of tau and amyloid proteins, neuroimaging techniques have significantly advanced precision medicine in AD. This allows for disease stratification and may even facilitate an earlier diagnosis [214,215].

The field of AD precision medicine has been significantly improved by the application of artificial intelligence. Artificial intelligence (AI) has transformed the processing, integration, and analysis of large-scale data produced by wearable sensors and “OMICS” or “EXPOsOMICS”-based measurement methodologies. AI includes machine learning (ML), deep learning (DL), and artificial neural networks (ANN). These AI systems are very good at finding patterns, assessing large volumes of data, and forecasting the likelihood of developing diseases. AI systems have been approved by the US Food and Drug Administration (FDA) for enhancing medical imaging assessments, including the identification of suspicious lesions [37,213,216]. These systems have demonstrated notable advances in diagnostic performance, particularly in radiology.

Combination Therapies

Combination therapies of the pharmacodynamic and pharmacokinetic types are available. Pharmacodynamic agent combinations for AD may consist of DMTs, which change the course of the disease, or symptomatic medicines, which reduce the symptoms without changing the course of the disease. As an illustration, ongoing clinical trials combine novel DMT- or symptomatic-based treatments with standard-of-care medications (such as memantine or cholinesterase inhibitors) [217,218,219]. Combination therapy is another example; it targets different pharmacologic targets (such as tau and amyloid) or targets pathways sequentially by using successive combinations. Another type of combination therapy is represented by multifunctional compounds, which incorporate several activities or targets into a single medication. One example of this is rasagiline, which combines neuroprotective and amyloid-processing properties with monoamine oxidase B inhibition [217].

It becomes a comprehensive care paradigm when pharmacological therapies are combined with lifestyle changes, cognitive training, and psychosocial support. This approach is adaptable enough to be customized for every single patient, ensuring optimal treatment outcomes [220,221].

## 5. Prospects for the Future of Phytochemical Treatment of AD

Here, we thoroughly examined herbal therapy approaches, which have long been a popular therapeutic choice for the treatment of AD. Finding preventative drugs derived from conventional herbal remedies is one approach to treating AD.

New advances in synthetic biology and genomics open up new avenues for discovering and using the medicinal properties of plants. The search for new leads in AD treatment could be fruitful if herbal remedies are evaluated for lead chemicals based on their physicochemical properties and predicted BBB actions.

Science and well-studied customs will once again take the stage as the world’s natural cures prepare to enter an exciting new chapter. Future research may focus on natural substances that have the capacity to cross the blood–brain barrier, extended therapeutic windows, distinct pharmacological goals, and minimal side effects.

## 6. Conclusions

AD is a neurological condition that worsens with time and mostly affects cognitive abilities in the brain. It causes disorientation, memory loss, behavioral abnormalities, and personality changes. 

AD is a debilitating disorder that is expected to become more frequent as our society ages. Only a small percentage of AD patients benefit from the two DMTs that are currently approved for therapy. Discovering novel drugs and therapies that not only treat the disease’s symptoms but also stop or reverse its progression is crucial. Fortunately, there are other DMTs being developed that might be approved by the FDA. It is important for practicing practitioners to understand the various therapeutic aims of these new drugs.

A DMT with a particular therapeutic target may be more effective than other DMTs with different targets due to the complex and all-encompassing pathogenesis of AD, particularly if the therapeutic target of the DMT being administered aligns with the most salient pathological findings of the subpopulation.

Nevertheless, several natural compounds have demonstrated potential for treating AD in both in vitro and in vivo studies. However, because humans and animals in studies differ physiologically, clinical trials are still necessary to verify the safety and efficacy of these drugs.

However, additional research comparing DMTs with diverse therapeutic targets and their effectiveness in treating various patient groups with AD is required for validation. Certain dysregulated proteostasis, dysregulated circadian rhythms, inflammation, oxidative stress, tau proteinopathies, and dysregulated metabolism may all be treated therapeutically with some DMTs. Some DMTs that are awaiting FDA approval focus on neuroprotection or increasing neuroplasticity in an effort to slow the progression of AD. In addition, in order to effectively control AD over the long term, thorough study on each herb’s extraction method, dosage, group, mechanism of action, effectiveness, etc., must be conducted in carefully planned clinical trials.

## Figures and Tables

**Figure 1 molecules-29-05131-f001:**
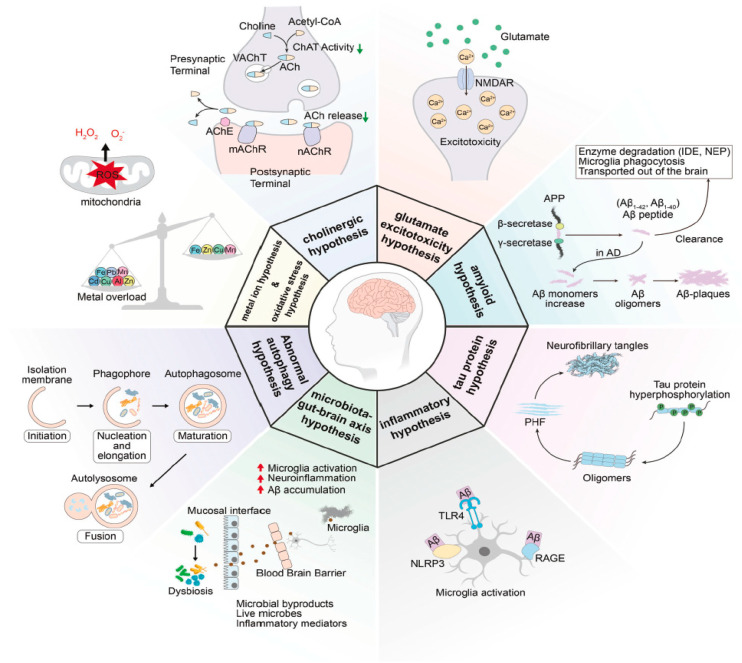
Pathophysiology of the AD hypothesis [8]. Reproduced with permission from Jifa Zhang, Yinglu Zhang, Jiaxing Wang, Yilin Xia, Jiaxian Zhang, and Lei Chen, in the journal *Signal Transduction and Targeted Therapy*; published by Springer Nature, 2024, licensed under CC BY 4.0.

**Figure 2 molecules-29-05131-f002:**
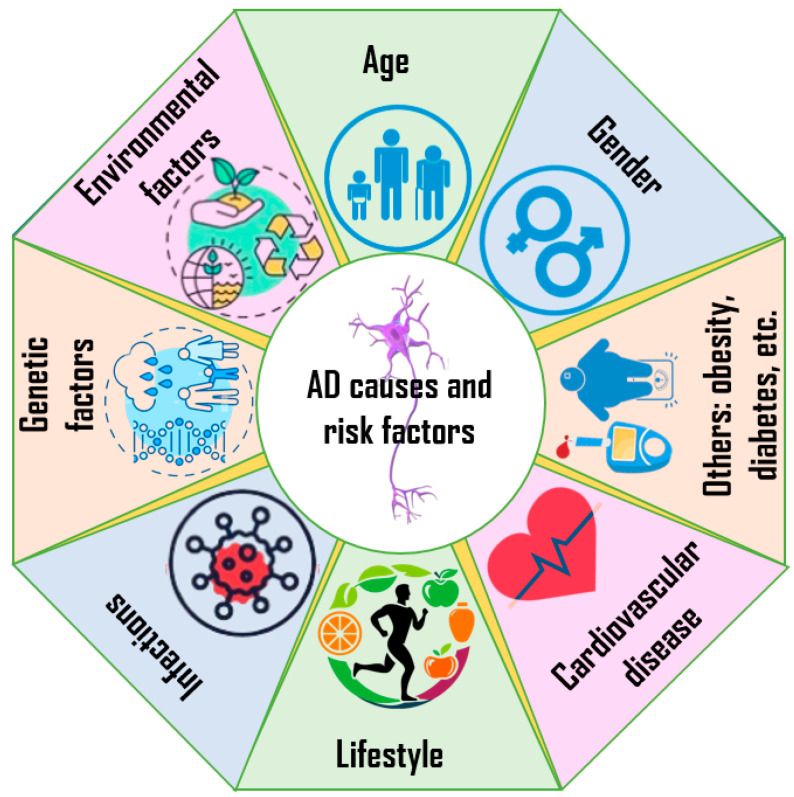
Causes and factors of AD.

**Figure 3 molecules-29-05131-f003:**
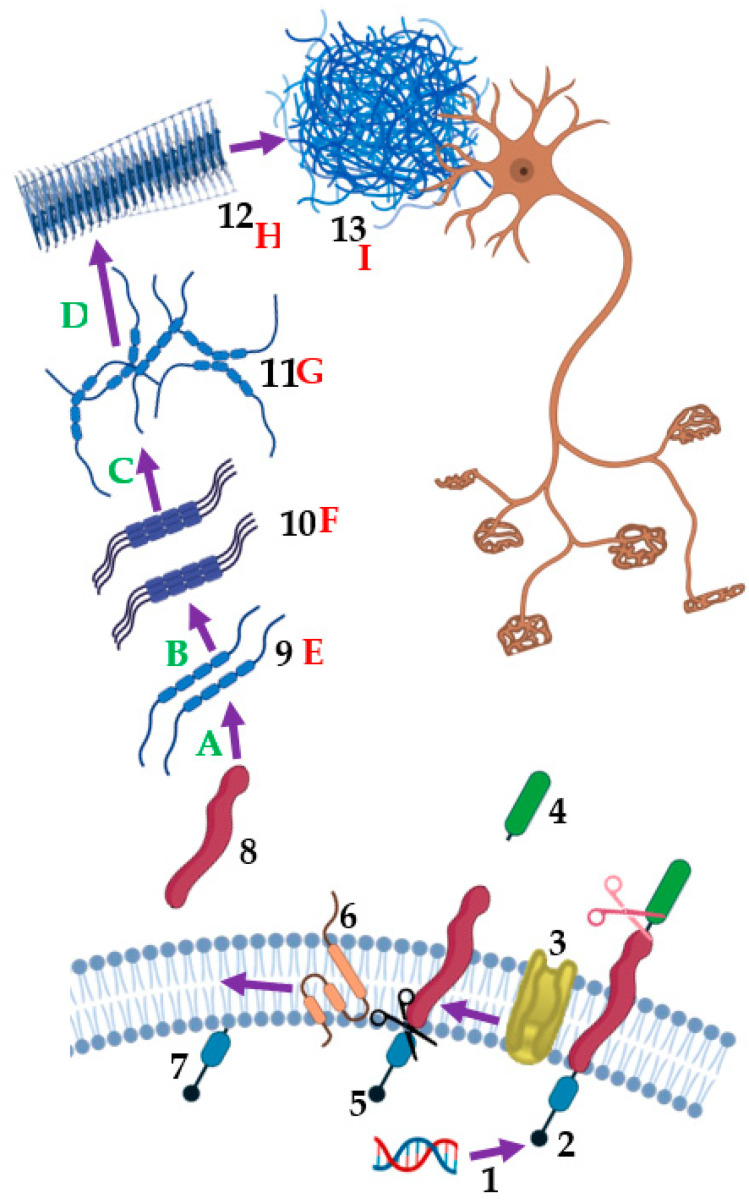
An example of how the major Aβ-targeted treatments work, based on the amyloid cascade theory. (1) RNA based therapies, (2) Amyloid precursor protein (APP), (3) β-Site amyloid precursor protein cleaving enzyme (BACE), (4) Soluble amyloid precursor protein-β (sAPPβ), (5) 99-Residue C-terminal fragment (C99), (6) γ-Secretase inhibitors/modulators, (7) Amyloid precursor protein intracellular domain (AICD), (8) Amyloid-β (Aβ), (9) Monomers, (10) Oligomers, (11) Protofibrils, (12) Fibrils, (13) Amyloid plaques. A to D (green) indicate the targets of Aβ aggregation inhibitors, E to I (red) indicate the targets of active and passive immunotherapies.

**Figure 4 molecules-29-05131-f004:**
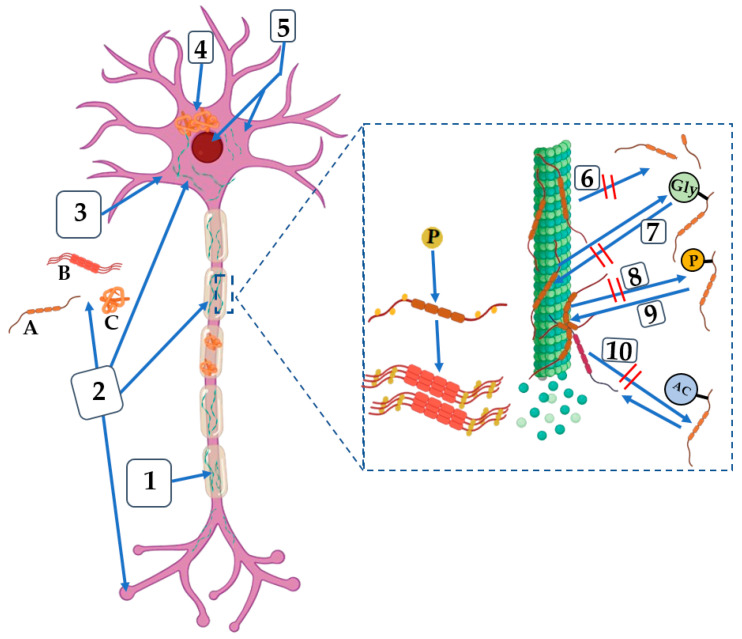
An example of the primary therapeutic targets connected to tau and their mode of action. (1) Microtubule stabilizing agent; (2) Active or passive immunization; (3) Modulation of autophagy or proteasomal digestion; (4) Aggregation inhibitor; (5) Tau expression inhibitor; (6) Caspase inhibitors; (7) O-GlcNAcase inhibitors (OGA); (8) Kinase inhibitors; (9) Phosphate activators; (10) Acetylation inhibitors. A, monomers; B, oligomers; C, aggregate; P, phosphate; AC, acetyl group; Gly, glycosyl group.

**Figure 5 molecules-29-05131-f005:**
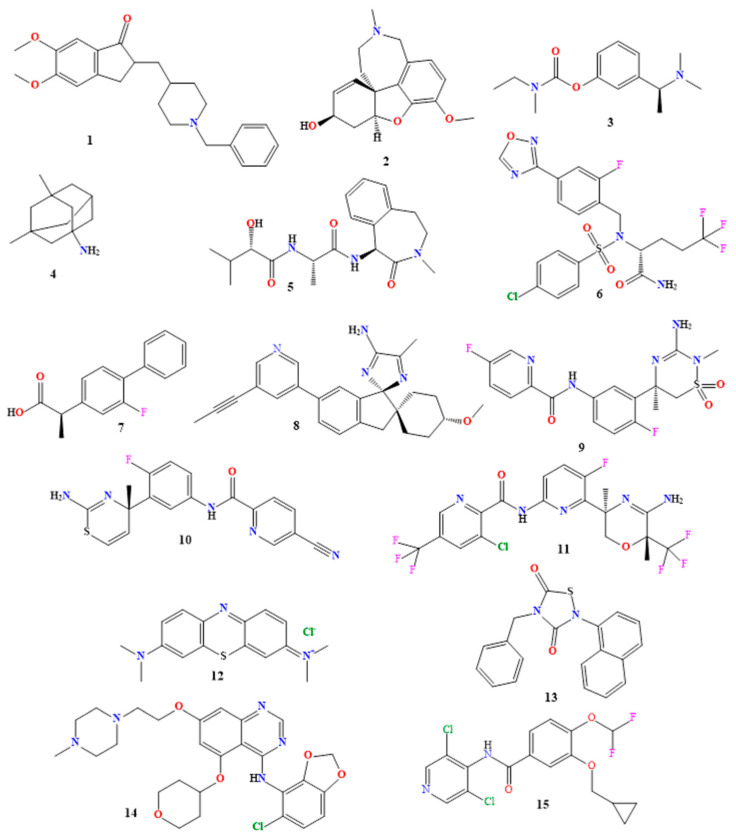
Chemical structures of (1) donepezil, (2) galantamine, (3) rivastigmine, (4) memantine, (5) semagacestat, (6) avagacestat, (7) tarenflurbil, (8) lanabecestat, (9) verubecestat, (10) atabecestat, (11) umibecestat, (12) methylene blue, (13) tideglusib, (14) saracatinib, and (15) roflumilast.

**Table 1 molecules-29-05131-t001:** AD signs and symptoms by stages.

Stage	Description	Signs/Symptoms
Mild (early)	The early stages are usually ignored because it is rare to see the signs and symptoms. Neuropsychiatric issues may be observed in the mild stage of AD, including depression, irritability, anxiety, and apathy [29,30,31].	Memory problems Linguistic disruption Losing or misplacing items
Moderate	The signs and symptoms become more pronounced and noticeable. These could last for a long time [24,30,32].	Forgetfulness of names and current events Growing challenges in communicating Sleep pattern alterations Wandering and asking a lot of questions
Severe (late)	It is on the verge of complete dependence and inaction. The signs and symptoms start to show more clearly. Every past proficiency eventually declines [29,33,34].	Need for assistance in performing daily duties and personal care Loss of capacity to control how they move Becoming bedridden, becoming silent, and incontinent Many difficulties and complications

**Table 2 molecules-29-05131-t002:** Clinical studies of inflammatory targeting, tau, and amyloid beta as disease-modifying biological agents.

Phase	Agent	Mechanism of Action	Target	Trial Code	Date
Phase I	ALN-APP	RNAi to reduce Aβ-related downstream events and APP	Amyloid beta	NCT05231785	2022–2025
	ALZ-101	Amyloid beta-directed vaccine		NCT05328115	2021–2023
	AV-1959	Anti-amyloid vaccine		NCT05642429	2023–2026
	Remtemetug	Monoclonal antibody (anti-amyloid)		NCT04451408	2020–2024
	SHR-1707	Monoclonal antibody (anti-amyloid)		NCT06114745	2024–2025
	APNmAb005	Anti-tau antibody	Tau	NCT05344989	2022–2024
	MK-2214	Anti-tau monoclonal antibody		NCT05466422	2022–2025
	NIO752	Anti-tau antisense oligonucleotide		NCT05469360	2023–2024
	Bacillus Calmette– Guerin	Vaccination to increase resistance to mechanisms related to AD	Inflammation	NCT06078891	2023–2024
	CpG1018	Reduced Aβ plaques and tau pathology due to toll-like receptor nine agonist		NCT05606341	2023–2024
	IBC-Ab002	Immunocheckpoint inhibitor against programmed death-ligand 1 (PD-L1)		NCT05551741	2023–2024
Phase II	ABBV-916	Anti-amyloid antibody	Amyloid beta	NCT05291234	2022–2030
	ACI-24.060	Immunizations against amyloid-beta protein are stimulated by vaccines		NCT05462106	2022–2026
	ALZN002	Autologous dendritic cells embedded with peptides of mutant amyloid beta		NCT05834296	2023–2028
	Lecanemab	Monoclonal antibody against amyloid that targets amyloid plaques and protofibrils		NCT01767311	2021–2025
	Trontinemab	Monoclonal antibody called gantenerumab, which targets plaques and oligomers		NCT04639050	2021–2027
	Bepranemab	Binding of a monoclonal antibody against tau to its core region	Tau	NCT04867616	2021–2024
	BIIB080	Tau mRNA translation into tau protein is inhibited by an antisense oligonucleotide		NCT05399888	2022–2027
	E2814	Monoclonal antibody (anti-tau)		NCT04971733	2021–2025
	JNJ-63733657	Monoclonal antibody targeted at soluble tau (mid-region of tau)		NCT04619420	2021–2025
	AL002	Monoclonal antibody targeting TREM2 receptors	Inflammation	NCT05744401	2023–2025
	Bacillus Calmette- Guerin	Vaccination to increase resistance to mechanisms related to AD		NCT05004688	2022–2023
	Canakinumab	Anti-IL-1-beta monoclonal antibody		NCT04795466	2021–2024
	Interleukin-2	Restore regulatory T-cell function		NCT06096090	2023–2025
	Pegipanermin	Neutralizes TNF-alpha		NCT05522387	2023–2026
	Pepinemab	Semaphorin 4D-specific monoclonal antibody selectively suppresses the release of inflammatory cytokines		NCT04381468	2021–2023
	Proleukin	IL-2 immunomodulator		NCT05468073	2022–2025
	Sargramostim	Hematopoietic growth anti-inflammatory factor; granulocyte macrophage colony-stimulating factor		NCT04902703	2022–2024
	TB006	Monoclonal antibody directed against galactose-specific lectin (galectin) 3, anti-inflammatory		NCT05476783	2022–2024
	Tdap	To promote inflammatory protection, acellular pertussis vaccination is given with decreased diphtheria toxoid and tetanus toxoid		NCT05183516	2023
Phase III	Aducanumab	Monoclonal antibody targeting oligomers and plaques that is anti-amyloid	Amyloid beta	NCT04241068 NCT05310071	2020–2023 2022–2025
	Donanemab	Anti-amyloid monoclonal antibody specific for pyroglutamate plaque amyloid		NCT04437511 NCT05026866 NCT05508789 NCT05738486	2020–2023 2021–2027 2022–2027 2023–2024
	Gantenerumab	Monoclonal antibody (anti-amyloid)		NCT01760005	2012–2027
	Lecanemab	Anti-amyloid monoclonal antibody directed at amyloid protofibrils and amyloid plaques		NCT01760005 NCT03887455 NCT04468659 NCT05269394	2012–2027 2019–2027 2020–2027 2021–2027
	Remternetug	Monoclonal antibody (anti-amyloid)		NCT05463731	2022–2025
	Solanezumab	Monoclonal antibody (anti-amyloid)		NCT01760005	2012–2027
	E2814	Anti-tau monoclonal antibody	Tau	NCT01760005 NCT05269394	2012–2027 2021–2027
	Semaglutide	GLP-1 agonist; anti-inflammatory	Inflammation	NCT04777396 NCT04777409 NCT05891496	2021–2025 2021–2025 2023–2024

**Table 3 molecules-29-05131-t003:** Clinical studies of inflammatory targeting, tau, and amyloid beta as disease-modifying small molecule agents.

Phase	Agent	Mechanism of Action	Target	Trial Code	Date
Phase I	BMS-984923	Silent allosteric modulator (SAM) of mGluR5	Amyloid beta	NCT05804383 NCT05817643	2023–2024 2023–2023
	OLX-07010	Inhibits tau self-aggregation	Tau	NCT05696483	2023–2024
	Emtricitabine	Nucleoside reverse transcriptase inhibitor (NRTI	Inflammation	NCT04500847	2021–2024
Phase II	APH-1105	Amyloid precursor protein secretase modulator; alpha-secretase modulator	Amyloid beta	NCT03806478	2023–2024
	MIB-626	Adenine dinucleotide stimulator sirtuin-nicotinamide to increase alpha-secretase		NCT05040321	2021–2024
	PRI-002	Interferes with A-beta 42 oligomerization to hinder its production and promote its decrease		NCT06182085	2023–2026
	Valiltramiprosate	Aggregation inhibitor		NCT04693520	2020–2024
	Varoglutamstat	Using glutaminyl cyclase (QC) enzyme inhibitor, the amount of pyroglutamate Aβ produced is decreased		NCT03919162 NCT04498650	2021–2023 2020–2024
	LY3372689	O-GlcN Acase enzyme inhibitor	Tau	NCT05063539	2021–2024
	Methylene Blue	Tau protein aggregation inhibitor		NCT02380573	2015–2023
	Baricitinib	Janus kinase (JAK) inhibitor	Inflammation	NCT05189106	2022–2024
	Dasatinib + Quercetin	Senescent cells can be eliminated by dasatinib by inducing apoptosis in them; quercetin is a flavonoid		NCT04685590 NCT04785300 NCT05422885	2021–2025 2022–2023 2022–2024
	Lenalidomide	Originally authorized for the treatment of multiple myeloma; anti-inflammatory and immunomodulatory		NCT04032626 NCT06177028	2020–2023 2024–2026
	L-Serine	Dietary amino acid that occurs naturally; prevents harmful misfolding		NCT03062449	2023–2024
	Montelukast buccal film	Leukotriene receptor antagonist (LTRA); has anti-inflammatory properties		NCT03402503	2018–2024
	PrimeC	Combined product that addresses RNA regulatory issues, iron buildup, and inflammation		NCT06185543	2023–2025
	Senicapoc	Calcium-activated potassium channel inhibitor		NCT04804241	2022–2024
	Valacyclovir	HSV-1 and -2 antiviral; lessens viral-related “seeding” of amyloid plaque deposition		NCT03282916	2028–2024
Phase III	Donanemab	Monoclonal antibody against plaque amyloid derived from pyroglutamate	Amyloid beta	NCT04437511 NCT05026866 NCT05508789 NCT05738486	2020–2023 2021–2027 2022–2027 2023–2024
	Valiltramiprosate	Tramiprostate prodrug		NCT04770220	2021–2024
	Masitinib	Tyrosine kinase inhibitor which reduces mast cell and microglia/macrophage activity, exhibiting neuroprotective effects	Inflammation	NCT05564169	2024–2026

## Data Availability

Not applicable.

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
