# Peer review of "Recent Advances in Therapeutics for the Treatment of Alzheimer’s Disease"

_molecules, 2024, doi:10.3390/molecules29215131_

Round 1

Reviewer 1 Report

Comments and Suggestions for Authors

Points of concern, discrepancy, and/or required clarification are as follows:

Abstract: the abstract wanders around, giving unnecessary focus on sympotomological drugs for AD (lines: 13-17; 20-22).  Refocus abstract to DMTs.

Section 1. Overview of Alzheimer’s disease is limited in concept and requires further organization therein.  The section provides a number of generalized concept statements thrown together.

§  Concepts and references should be further vetted.  Many references simply do not accurately or adequately align with the associated statement/concepts addressed.  This is an issue throughout the manuscript.  Examples:

·         Line 38; ref 1-7 only loosely apply to statement… focus only appropriate refs.

·         Line 40; Ref 8-12-  review refs as they do not appear to apply to the actual statement referenced. The actual sentence “Because…” could be eliminated, as it is overly open-ended.  The whole paragraph (lines 54-58) is a random non-connecting statements.

·         Lines 54-56, is overly generalized, and inaccurate as written… “because of neuronal death and atrophy in the temporofrontal cortex, Alzheimer’s disease causes…”  this is not the order of effects.

§  No address of hippocampus, Amyloid-beta oligomers, etc.

§  Figure-1 needs to be reworked.  Generalized concepts without cohesion of concept.  Also, the figure states (blue oval): “cholinergic hypothesis/ Reduction in Acetylcholinesterase”….  it is it not a reduction in “acetylcholinesterase” that is the issue with AD, just the opposite.  Carefully review figure.

Section 2.

§  over repetition of concepts, with limited depth.

§  Some information would be better suited to the Overview section.

§  Again, questionable reference citations.  Ex. Statement in lines 216-220 notes references 126, 127… but these do not align with statement. 

Section 3.

§  While addressing DMTs on several levels, this section still goes back into symptomological focused drugs, as well as non-pharmacological interventions, which are not to be the focus on the review.

 Minor points:

o   The manuscript assigns the acronym of “AD” to Alzheimer’s disease, but does not consistently use it (i.e. uses Alzheimer’s or Alzheimer’s disease).  Once assigned it should be used throughout the entirety of the manuscript.  

o   Reduction of extraneous wording would benefit the manuscript.  Example: “As previously said” (line 47) is unnecessary.

Summary

Overall, the review has overly generalized information and keeps going back to Sympological focused medication (i.e. not the focus of the review as titled), with a good deal of redundancy with other recent reviews (e.g. Cummings J. et al., 2024, Alzheimer’s disease drug development pipeline: 2024).  The review requires further organization and detailed delineation of actual new DMT listed in the tables 2-3.  Reference sourcing is a concern throughout manuscript. References need to fully align accurately and appropriate with the associated statement, without excess use of other reviews as references. Original/primary references for address of actual data should be used.

Comments on the Quality of English Language

English quality is good.  However, the structure and flow of sentences within paragraphs needs further enhancement.

Author Response

Reviewer 1:

We thank the reviewer for his/her fruitful comments and suggestions.

Points of concern, discrepancy, and/or required clarification are as follows:

Comment: Abstract: the abstract wanders around, giving unnecessary focus on sympotomological drugs for AD (lines: 13-17; 20-22).  Refocus abstract to DMTs.

Response: To properly reflect the text, the abstract was revised and the title of the paper was altered.

Comment: Section 1. Overview of Alzheimer’s disease is limited in concept and requires further organization therein.  The section provides a number of generalized concept statements thrown together.

  • Concepts and references should be further vetted.  Many references simply do not accurately or adequately align with the associated statement/concepts addressed.  This is an issue throughout the manuscript.  Examples:
  • Line 38; ref 1-7 only loosely apply to statement… focus only appropriate refs.
  • Line 40; Ref 8-12- review refs as they do not appear to apply to the actual statement referenced. The actual sentence “Because…” could be eliminated, as it is overly open-ended.  The whole paragraph (lines 54-58) is a random non-connecting statement.
  • Lines 54-56, is overly generalized, and inaccurate as written… “because of neuronal death and atrophy in the temporofrontal cortex, Alzheimer’s disease causes…”  this is not the order of effects.
  • No address of hippocampus, Amyloid-beta oligomers, etc.
  • Figure-1 needs to be reworked.  Generalized concepts without cohesion of concept.  Also, the figure states (blue oval): “cholinergic hypothesis/ Reduction in Acetylcholinesterase” ….  it is it not a reduction in “acetylcholinesterase” that is the issue with AD, just the opposite.  Carefully review figure.

Response: As advised, the introduction was changed (Turquoise).

Section 1.1 has been changed and amended in accordance with the reviewer's recommendations. Figure 1 was modified to make it more precise and thorough. References have all been reviewed and modified as necessary. All changes are marked with a yellow color.

To accommodate the reviewers' remarks and recommendations, Figure 2 was included. 

Comment: Section 2.

  • over repetition of concepts, with limited depth.
  • Some information would be better suited to the Overview section.
  • Again, questionable reference citations.  Ex. Statement in lines 216-220 notes references 126, 127… but these do not align with statement. 

Response: The updated version takes into account every comment. The references were reviewed and fixed, and repeated sentences were removed. Text changes are indicated in gray.

Comment: Section 3.

  • While addressing DMTs on several levels, this section still goes back into symptomological focused drugs, as well as non-pharmacological interventions, which are not to be the focus on the review.

Response: The title of the review was modified to encompass all current AD treatments, with a focus on the use of DMTs. -There is no longer a section on non-pharmacological interventions.

 Minor points:

Comment: The manuscript assigns the acronym of “AD” to Alzheimer’s disease, but does not consistently use it (i.e. uses Alzheimer’s or Alzheimer’s disease).  Once assigned it should be used throughout the entirety of the manuscript.

Response: (Done, marked in turquoise color)

Comment: Reduction of extraneous wording would benefit the manuscript.  Example: “As previously said” (line 47) is unnecessary.

Response: (Done)

Comment: Summary

Overall, the review has overly generalized information and keeps going back to Sympological focused medication (i.e. not the focus of the review as titled), with a good deal of redundancy with other recent reviews (e.g. Cummings J. et al., 2024, Alzheimer’s disease drug development pipeline: 2024). 

Response: After reviewing a number of studies, we were able to collect the tables listed below, which we verified using the ClinicalTrials.gov website:

     -Amirrad, Farideh, et al. "Alzheimer’s Disease: Dawn of a New Era?." (2017).

     -Cummings, Jeffrey, et al. "Alzheimer's disease drug development pipeline: 2020, 2021, 2022, 2023 and 2024." Alzheimer's & dementia: translational research & clinical interventions

      -Prajapati, Vipul, et al. "New biologicals and biomaterials in the therapy of Alzheimer's disease." Alzheimer's Disease and Advanced Drug Delivery Strategies. Academic Press, 2024. 93-114.

      - Piton, Morgane, et al. "Alzheimer’s disease: advances in drug development." Journal of Alzheimer's Disease 65.1 (2018): 3-13.

The following references were cited before the tables [38, 46, 68, 70, 71, 157, 182, 183, 189].

Comment: The review requires further organization and detailed delineation of actual new DMT listed in the tables 2-3.  Reference sourcing is a concern throughout manuscript.

Response: Each particular trial is identified by the cods in the fifth column of Tables 2 and 3, which also include the sponsor, development laboratory, and responsible party. Since most trials have not been the subject of published research, please see the US government's ClinicalTrials.gov website. Additionally, the studies were previously cited in published articles using codes alone, as demonstrated in the published manuscripts below:     -Amirrad, Farideh, et al. "Alzheimer’s Disease: Dawn of a New Era?." (2017).

     -Cummings, Jeffrey, et al. "Alzheimer's disease drug development pipeline: 2020, 2021, 2022, 2023 and 2024." Alzheimer's & dementia: translational research & clinical interventions

      -Prajapati, Vipul, et al. "New biologicals and biomaterials in the therapy of Alzheimer's disease." Alzheimer's Disease and Advanced Drug Delivery Strategies. Academic Press, 2024. 93-114.

      - Piton, Morgane, et al. "Alzheimer’s disease: advances in drug development." Journal of Alzheimer's Disease 65.1 (2018): 3-13.

Comment: References need to fully align accurately and appropriately with the associated statement, without excess use of other reviews as references. Original/primary references for the address of actual data should be used.

Response: Every reference was examined, and only authentic so

Reviewer 2 Report

Comments and Suggestions for Authors

The review entitled "Recent Advances in Disease-modifying therapeutics (DMT) for the Treatment of Alzheimer's Disease" provides a  summary of current medication used in clincial and currently tested in different phases of clinical trials. The manuscript is well written. 

I have a couple of comments which should be fixed prior publication regarding:

Section 1.2

The section below should be re written in a more accurate way (author should read other review on pathophysiology of AD) and place in section 1.1

“Another clue that someone has AD is the existence of neurofibrillary tangles, which 101 are primarily composed of hyperphosphorylated tau protein. Communication is hampered and cell death is encouraged by tau aggregation' impairment of the neuronal 103 transport system [76–79]. Additionally, an accumulation of beta-amyloid peptides in the 104 brain leads to plaque formation, which impairs cell function and triggers inflammatory 105 responses that kill and destroy neurons [80–84]. Additionally, aging is the single biggest 106 risk factor for the development of AD since it impairs many biological processes, such as 107 neurogenesis, synapse function, and mechanisms for cell repair. Moreover, the accumu- 108 lation of amyloid plaques and neurofibrillary tangles, which can cause cognitive decline, 109 increases the vulnerability of the brain [24, 54, 84-89].”

Section 2.3.1 more specific information about the type of medication discussing on Abeta (such as monoclonal antibody) which is mentioned later on.

Minor comment: abbreviation should be consistent across the full manuscript

Author Response

Reviewer 2:

We thank the reviewer for his/her fruitful comments and suggestions.

Comment: The section below should be re written in a more accurate way (author should read other review on pathophysiology of AD) and place in section 1.1

Another clue that someone has AD is the existence of neurofibrillary tangles, which 101 are primarily composed of hyperphosphorylated tau protein. Communication is hampered and cell death is encouraged by tau aggregation' impairment of the neuronal 103 transport system [76–79]. Additionally, an accumulation of beta-amyloid peptides in the 104 brain leads to plaque formation, which impairs cell function and triggers inflammatory 105 responses that kill and destroy neurons [80–84]. Additionally, aging is the single biggest 106 risk factor for the development of AD since it impairs many biological processes, such as 107 neurogenesis, synapse function, and mechanisms for cell repair. Moreover, the accumu- 108 lation of amyloid plaques and neurofibrillary tangles, which can cause cognitive decline, 109 increases the vulnerability of the brain [24, 54, 84-89].”

Response: The entire section was revised in light of the reviewer's suggestions and comments.

Comment: Section 2.3.1 more specific information about the type of medication discussing on Abeta (such as monoclonal antibody) which is mentioned later on.

Response: Thank you very much for this helpful recommendation. Section 3 "AD Current Treatment Drugs" discusses monoclonal antibodies as well as other topics.

Comment: Minor comment: abbreviation should be consistent across the full manuscript (Done)

Response: Non-Pharmacological Interventions was removed as a sub-section. The entire text now uses the same abbreviations.

Reviewer 3 Report

Comments and Suggestions for Authors

1.       Please rewrite the statements in line 47 – 49. Please clarify the cause-and-effect relationship.

2.       Please add a figure to describe the causes of AD in detail.

3.       Please add subsection to describe the pathophysiological mechanisms of AD.

4.       Please add a figure to describe the pathophysiological mechanisms of AD and related drugs.

5.       Please delete the paragraph of Non-Pharmacological Interventions.

6.       Please check the abbreviations of Alzheimer’s disease (AD) and disease-modifying therapies (DMTs) in the full text.

Author Response

Reviewer 3:

We thank the reviewer for his/her fruitful comments and suggestions.

Comment: Please rewrite the statements in line 47 – 49. Please clarify the cause-and-effect relationship.

Response: Done.

Comment: Please add a figure to describe the causes of AD in detail.

Response: Done, figure 2 was added.

Comment: Please add subsection to describe the pathophysiological mechanisms of AD

Response: As indicated in yellow, the requested subsection was added to section 1.1.

Comment: Please add a figure to describe the pathophysiological mechanisms of AD and related drugs

Response: Figure 1 was added, figure 1 – the drugs were indicated in another section.

Comment: Please delete the paragraph of Non-Pharmacological Interventions Response: The paragraph was deleted.

Comment: Abbreviations of Alzheimer’s disease (AD) and disease-modifying therapies (DMTs) in the full text.

Response: Done. The abbreviations are marked in turquoise.